# Reaction blueprints and logical control flow for parallelized chiral synthesis in the Chemputer

Mindaugas Šiaučiulis ◎ , Christian Knittl-Frank ◎ , S. Hessam M. Mehr ◎ , Emma Clarke & Leroy Cronin ◎ ✉

Despite recent proliferation of programmable robotic chemistry hardware, current chemical programming ontologies lack essential structured programming constructs like variables, functions, and loops. Herein we present an integration of these concepts into χDL, a universal high-level chemical programming language executable in the Chemputer. To achieve this, we introduce reaction blueprints as a chemical analog to functions in computer science, allowing to apply sets of synthesis operations to different reagents and conditions. We further expand χDL with logical operation queues and iteration via pattern matching. The combination of these new features allows encoding of chemical syntheses in generalized, reproducible, and parallelized digital workflows rather than opaque and entangled single-step operations. This is showcased by synthesizing chiral diarylprolinol catalysts and subsequently utilizing them in various synthetic transformations (13 separate automated runs affording 3 organocatalysts and 12 distinct enantioenriched products in 42–97% yield, up to > 99:1 er), including automated catalyst recycling and reuse.

Modern synthetic organic chemistry is rapidly incorporating an increasing number of digital, data-driven tools to accelerate discovery and the rate of production of small molecule targets[1], which is often a bottleneck in the discovery of new pharmaceuticals and materials[2,3]. However, despite significant recent advances in expanding the capabilities of automated synthesis platforms[4], improving their accessibility and versatility, synthetic organic chemistry remains a highly skilled, mostly manual endeavor. Only a limited amount of very specific areas of synthesis have benefitted from well-established automation protocols, such as synthesis of peptides[5], oligosaccharides[6], and oligonucleotides[7]. Such approaches rely on a small subset of exceptionally efficient chemical transformations, which can be repeatedly executed to consecutively grow the molecule via stepwise building-block addition, generally utilizing solid-supported reagents. On the contrary, automated liquid-phase reactions are often performed as high-throughput screens of single-step reactions[8], and multi-step preparative syntheses have not seen as much progress[9–11]. The few available automation platforms are often designed, built, and optimized for specific workflows, and significant redevelopment is required in order to access new classes of synthetic targets[12]. Some of the modern solutions to automated chemical synthesis span a range of approaches from standardized reagent-capsule-based synthesis machines[13], radial configuration[14] or rapidly reconfigurable plug-and-play flow chemistry platforms[15,16], to free-roaming dexterous robots which are able to perform operations with standard laboratory equipment[17]. Along with the advancements in the field of automation, the complexity of experimental procedures is increasing and is starting to leverage concepts from conventional programming languages[18–21]. However, no open universal standard for structured high-level chemical programming has been reported until now.

Complementary to the advances in automation, a growing number of synthetic methodologies targeted directly towards automated synthesis are being developed. Particularly, several novel iterative approaches have been recently reported, which enable a small set of

Advanced Research Centre, University of Glasgow, 11 Chapel Lane, Glasgow, UK. ✉e-mail: lee.cronin@glasgow.ac.uk

operations to be developed and optimized for automated execution (Fig. 1A). For example, MIDA[22] and TIDA[23] boronates have been developed as building blocks for C(sp²)−C(sp²) and C(sp²)−C(sp³) coupling reactions, utilizing their unique binary elution properties on silica to enable the inclusion of a standardized purification step within the synthetic workflow (Fig. 1B). In a complementary fashion, an automated C(sp³)−C(sp³) coupling strategy utilizing 1,2-metallate rearrangements of boronates to achieve iterative homologations with chiral carbenoid precursor building blocks (Fig. 1C) has recently been reported[24].

However, the preparation of the required building blocks and the functionalization of the resulting products still inevitably relies on conventional synthetic transformations, which pose a considerable challenge for automation. In particular, the preparation of chiral building blocks introduces additional synthetic challenges and time requirements, as synthetic procedures for stereoselective syntheses are usually different from their asymmetric counterparts, and selectively accessing a range of different stereoisomers requires multiple repetitions of synthetic work. Moreover, preparation of chiral building blocks often requires synthesis of the chiral catalysts, which is time-consuming. We envisioned that such crucial, yet repetitive and non-novel research tasks could be relegated to automated synthesis platforms, allowing the researchers to focus their efforts on other research tasks. Ideally, such synthesis would be digitally captured in a general template-like form, which would enable access to a range of products by simply changing the input parameters. Furthermore, this high-level encoding should be compatible with one-off specialized synthesis protocols in order to maintain universality.

We have previously introduced the Chemputer[25] as a modular automated synthesis platform (Fig. 2, top), and the Chemical Description Language (χDL)[26,27] as a universal way to digitally capture synthetic procedures (Fig. 2, bottom), which facilitates their automation, improves reproducibility, and enables construction of chemical databases for data-driven applications. These developments have allowed a facile translation of previously reported single-step reaction protocols to a robotically executable digital code. However, faithful translation of previously developed manual processes does not harness the full potential of programmable automated systems and new approaches are needed to truly digitize and modernize synthetic chemistry.

Herein, we report synthetic applications of expanded functionalities of χDL, such as blueprints, iteration, and parallel execution, all of which leverage the concepts well explored and thoroughly utilized in conventional programming languages. This constitutes a paradigm shift in how synthetic procedures are described, and promotes the development of more complex, digitally encoded protocols, which can be efficiently executed by programmable synthesis robots. Crucially, this enables automation of processes that would be very difficult or impossible to execute for a human chemist within standard working hours. Such approach is exemplified by a multistep, fully autonomous synthesis of several Hayashi-Jørgensen type organocatalysts, which have emerged as a particularly useful chiral catalyst scaffold[28], as well as their automated use, and recycling for iterative reuse. By harnessing the power of automation and digital, template-like synthesis code description, we can access a range of chiral building blocks and products in automation with improved convenience.

## Results and discussion

The synthesis of diarylprolinol silyl ether derivatives generally follows the same general sequence (Fig. 3, top left)[29]. Starting with a *N*-protected proline ester, consecutive organometallic addition of a Grignard reagent, *N*-deprotection, and *O*-protection or other derivatization affords the desired organocatalyst. Introduction of different aryl groups and silyl protecting groups allows for fine tuning of steric and electronic properties of the resulting catalyst, which controls their reactivity, setereocontrol, stability, as well as physical properties. We envisioned this class of catalysts as a prime example in how advances in programmable chemistry can significantly improve the synthetic workflow, enabling a general procedure to be used to conveniently synthesize a small in-house library of different catalysts for further studies. Owing to the generalizable nature of the reaction sequence, each of the reaction protocols can be easily encoded within a reaction Blueprint (Fig. 3, bottom left). Within a reaction Blueprint, the synthesis protocol is encoded in a highly general form, and any possible variations are explicitly detailed through the definition of input Reagents and Parameters.

Once a general procedure, or a Blueprint, is digitally encoded, the synthesis of a different catalyst requires only changing the definition of starting material and its physical properties, such as density and molecular weight (Fig. 3, top right). Essential reaction parameters, such as the reaction time for the in situ formation of the Grignard reagent, can easily be adjusted through the use of Parameters, although default values from the original general protocol may also be used. Importantly, digitally encoding the reaction procedure within a reaction

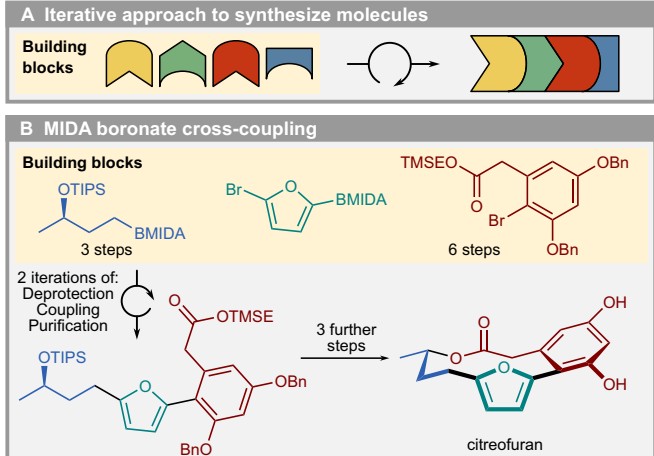
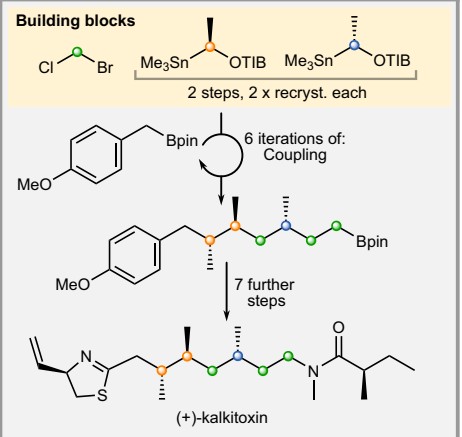

**Fig. 1 | Iterative approaches in synthesis and application in automation.**
**A** Iterative building block approaches make use of very selective and efficient transformations. High selectivity and efficiency is generally imparted by specific functional groups in the building blocks. **B** Iterative MIDA/TIDA boronate chemistry. Use of building blocks containing halide and boronate ester enables selective coupling, purification over multiple cycles. **C** Iterative homologations enable chain-growth of target molecules. The product retains the required synthetic handle, boronic ester, for further coupling cycles.

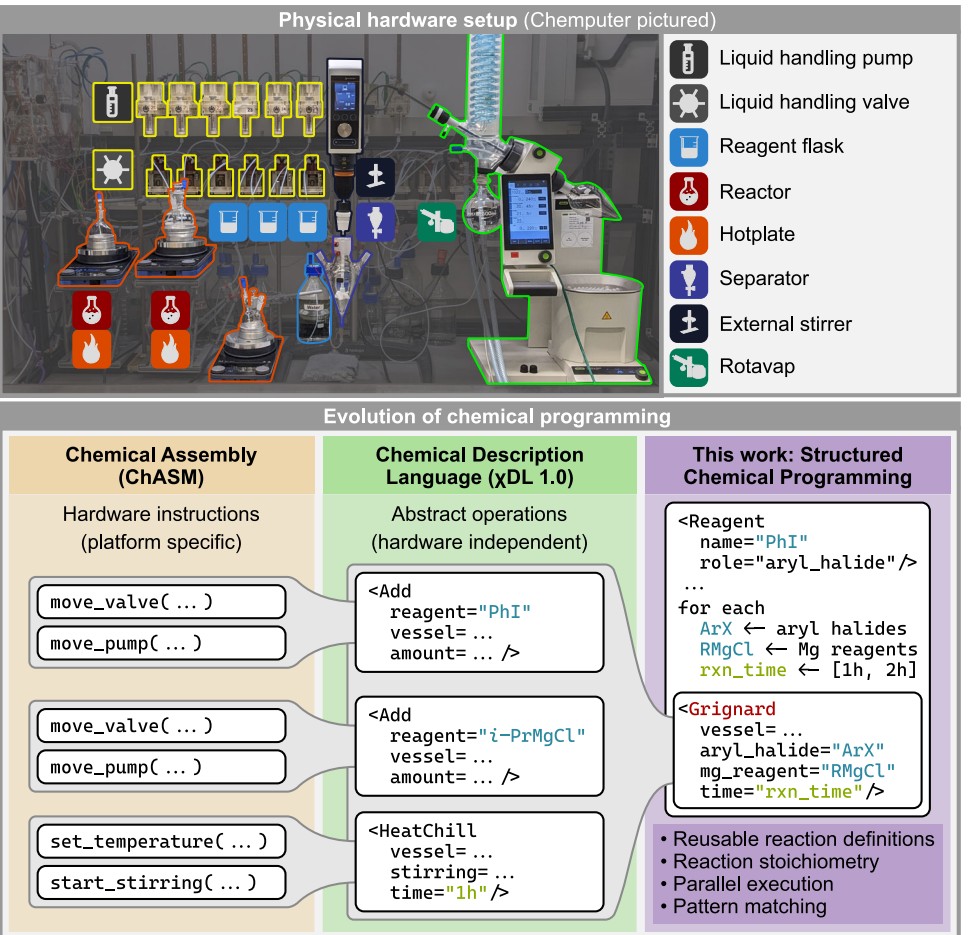

**Fig. 2 | The Chemputer and the evolution of chemical programming.** Top: The Chemputer (pictured) as a modular automated synthesis platform. Each module closely mimics standard laboratory equipment. Bottom: Evolution of chemical programming: (1) low-level hardware commands specific to implementation of hardware API; (2) chemical description language (χDL) as a hardware-independent abstract syntax for laboratory operations; (3) structured chemical programming utilizes concepts from conventional programming languages, such as reusable functions in the form of reactions Blueprints, iterations with pattern matching, execution scheduling. High-level syntax enables succinct encoding of complex chemical experiments.

Blueprint allows for facile sharing of digital synthetic code which does not require the end user to know intricate details of the synthetic procedure, and only requires a small amount of input parameters to be provided (Fig. 3, bottom right).

We sought to exemplify this approach by executing a general three-step sequence on our robotic platform to afford a small library of Hayashi-Jørgensen type organocatalysts **Cat-1–3** (Fig. 4). In the first step, the required organometallic reagent is prepared in situ by the robotic platform from the corresponding aryl halide starting material to afford diarylprolinol intermediates. Even though the formation and the reactivity of Grignard reagents generally follow a very comparable reaction protocol with a range of starting materials, minute variations in crucial reaction parameters, such as Grignard formation time, temperature, and nature of the magnesium reagent, are vital to the success of the process[30]. Capturing the general reaction, workup, and isolation protocol within a Blueprint, and allowing for modification of the crucial reaction parameters, enables a convenient and general digital encoding of the procedure for future use. Moreover, capturing each of the reaction steps as a well-defined blueprint with only specific points of modification allows for a convenient variation of input parameters during process development or reaction optimization. For example, in the synthesis of **(S)-Cat-1**, trifluoroacetic acid was used for the deprotection of the *N*-Boc intermediate and the same process was then used in the synthesis of **(S)-Cat-2** and **(S)-Cat-3**, unfortunately, in both cases either formation of an undesired side-product (see

Table S1) or decomposition of the intermediates was observed, leading to complex mixtures. Replacing the acid used in the deprotection to hydrogen chloride resulted in clean deprotection and, crucially, could be achieved by simply replacing a single input parameter (through further process optimization it was found that a modified workup was beneficial for efficacy of the process, and although not strictly necessary, the two deprotection protocols were separated into separate Blueprints for clarity).

Finally, a common silylation Blueprint was used for all of the syntheses, affording prolinol silyl ethers **(S)-Cat-1–3** in 58%, 77%, and 46% yields, respectively, over uninterrupted three-step synthetic sequence in automation. Crucially, no calculations or hard-coded reagent addition mass or volume parameters had to be changed between the runs – the reaction blueprints were encoded using relative stoichiometries and all reagent volume calculations were performed by the interpreter using available Reagent properties. The robotic platform required essentially no hardware reconfiguration between the syntheses, other than exchanging the input of aryl halide starting material used to generate the desired aryl Grignard reagents and the acid reagent used. Excitingly, the uninterrupted three-step sequence executed in automation afforded the final catalysts in yields comparable to those achieved by an expert chemist following the same procedure manually, for example, manual synthesis of *rac*-**Cat-2** afforded the catalyst in 83% yield (cf. 77% for automated synthesis of **(S)-Cat-2**). The execution of the whole reaction sequence took 34–38 hours of

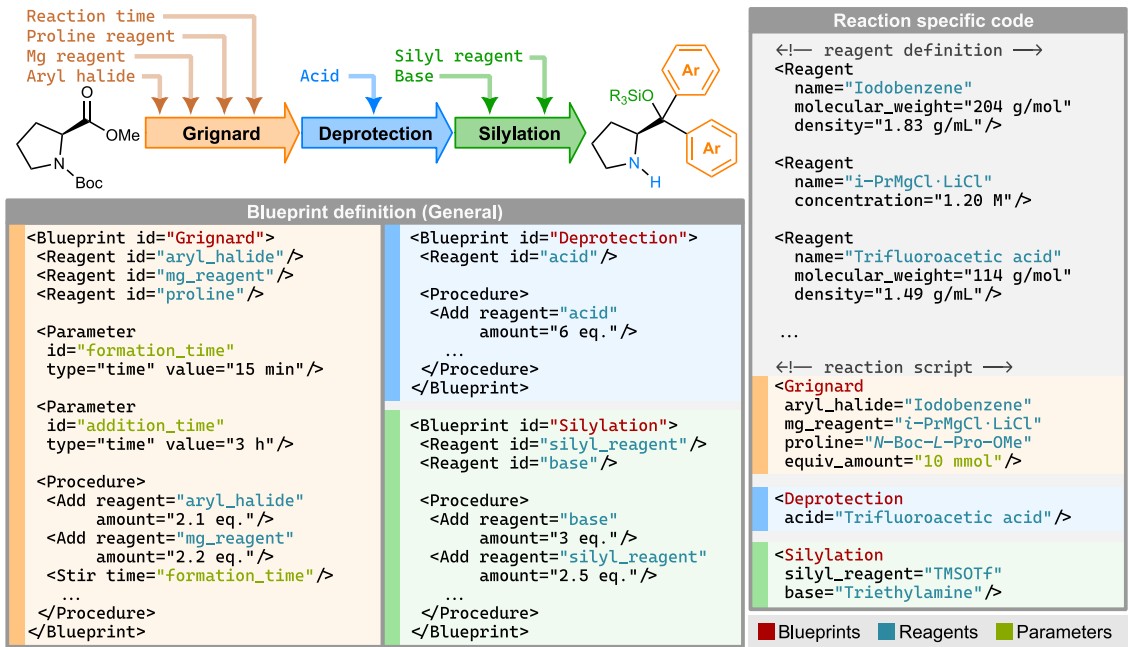

**Fig. 3 | Segmentation of the diarylprolinol silyl ether catalyst synthesis into reaction Blueprints.** Top left: General reaction scheme for preparation of diarylprolinol silyl ether catalysts through organometallic addition, Boc-deprotection, and silyl protection steps. A small number of input parameters can be changed in an otherwise general synthetic procedure. Bottom left: χDL Blueprints for each of the steps, using general reagent names, encoding default parameters, and reaction stoichiometry. Each of the modifiable process parameters is encoded as a variable Parameter within the Blueprint. Right: Synthesis of catalysts involves calling the Blueprints with the exact reagents required for the synthesis, reaction scale, and other variable reaction parameters. All required reagent properties are passed from the reagent definitions into the Blueprints for the calculations of exact masses and volumes to be added by the robotic platform, maintaining the generality of the Blueprint.

continuous and fully autonomous operation. Overall, the automated process could conveniently provide multi-gram quantities (2.1–3.5 g) of the desired catalysts **(S)-Cat-1–3** for further investigations.

Having developed a protocol for an on-demand automated synthesis of multi-gram quantities of the organocatalysts **(S)-Cat-1–3**, we attempted to use them in our robotic automated synthesis platform for a number of chemical transformations, utilizing both iminium and enamine modes of activation to obtain a range of chiral building blocks (Fig. 5). Previously reported reactions were performed without any further optimization to the reaction conditions.

An asymmetric Michael addition of activated isovaleraldehyde or valeraldehyde to methyl vinyl ketone afforded the desired dicarbonyl products **1a,b**[31]. An organocatalytic enantioselective α-chlorination of aldehydes has been reported to be sensitive to the rate of addition of the chlorinating agent, and has to be quenched immediately upon the completion of addition[32]. Chlorination of hydrocinnamaldehyde and the automated reductive quench of the reaction afforded the corresponding α-chloroalcohol **2**. Cinnamaldehyde **3** could be engaged in a range of transformations: stereoselective epoxidation with catalyst **(S)-Cat-1** afforded the β-hydroxy epoxide **4** after in situ reduction[33], and decarboxylative Michael addition afforded heteroaryl derivative **5** after in situ oxidation to the corresponding ester[34]. Both derivatizations were performed as part of the automated reaction process without intervention from a human chemist.

An organocatalyzed asymmetric Henry reaction between cinnamaldehyde and nitromethane using **(S)-Cat-2** afforded the desired nitroalkane product **6**[35], while the same materials under different reaction conditions and reversed stoichiometry (excess of the aldehyde) provided a domino iminium-iminium-enamine activation sequence product **7**[36]. The two diastereoisomers obtained could be separated by column chromatography and afforded the corresponding 3,4,5-trisubstituted cyclohexenes **7a,b** with excellent enantioselectivity. Importantly, the latter reactions do not proceed to full conversion with the more electron-deficient catalyst **(S)-Cat-1**. This highlights the necessity to have quick and reliable access to an in-house library of structurally related catalysts, enabled by a reproducible and automated workflow that can run on demand, in the background of other research.

Longer, multi-step cascade sequences could also be carried out – starting with an asymmetric epoxidation of heptenal with catalyst **(S)-Cat-1**, a four-reaction cascade process[37] afforded dihydrobenzofuran **8**, containing 3 contiguous stereo centers with excellent diasteo- and enantioselectivities, and the whole process was executed over 54 hours of continuous runtime. Crucially, the latter procedure demonstrates the significant advantages of automated workflows where multi-step procedures can be executed completely autonomously over a span of multiple days without the human chemist having to oversee or interact with the system until the process is fully complete.

Catalyst **(S)-Cat-3** has been previously developed as a water-soluble and recyclable version of the Hayashi-Jørgensen organocatalysts[38]. Although under the previously reported reaction conditions the catalyst **(S)-Cat-3** is able to maintain activity for at least four reaction cycles when utilized in an asymmetric Michael reaction between aldehydes and nitrostyrenes, the procedure is time-sensitive as leaving the catalyst in the mildly acidic aqueous reaction medium would lead to decomposition and loss of activity. Under the previously reported optimal conditions, five-hour reaction times were used to maintain the catalyst activity through several reaction cycles. Such strict timing requirements make the process highly inconvenient for the execution by human chemists. We envisioned that the process could be significantly improved by harnessing the advantages of automation. A general reaction procedure was encoded in a reaction Blueprint, which could then be executed multiple times in succession. Owing to each reaction being defined as a callable Blueprint, the same procedure can be conveniently reused with different inputs, akin to calling a function with different parameters.

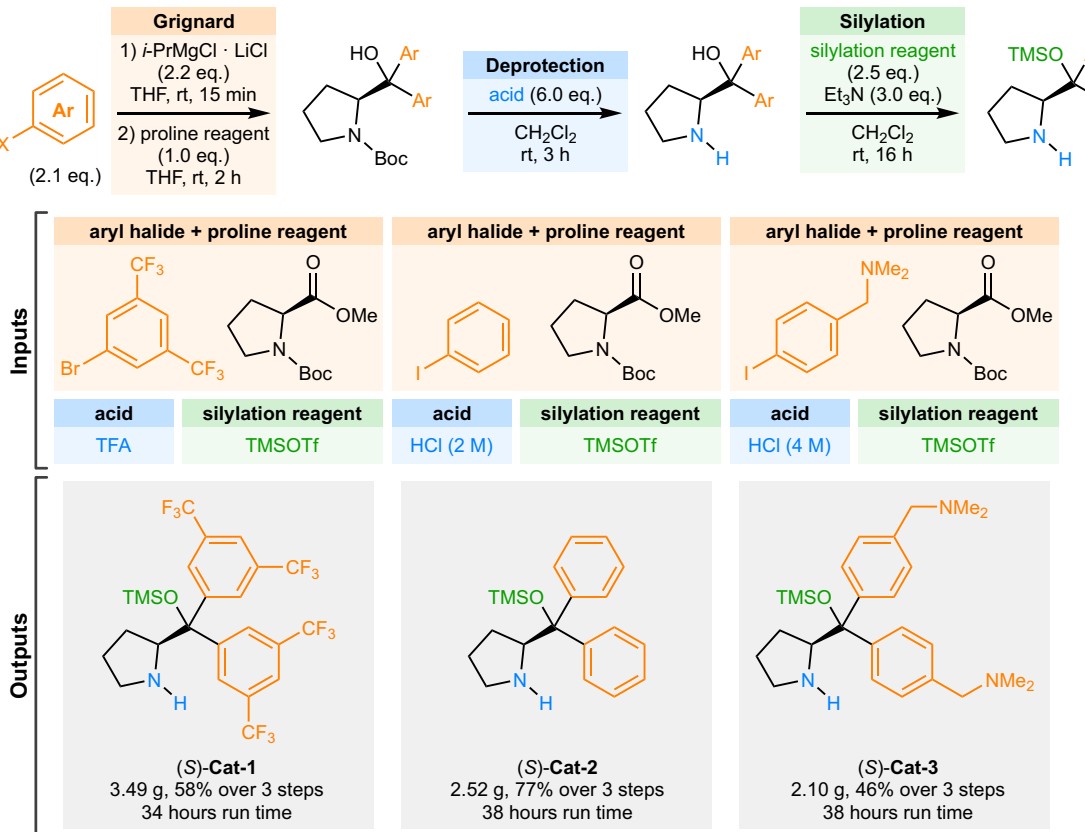

**Fig. 4 | Fully automated synthesis of catalysts (S)-Cat-1–3 in the Chemputer over uninterrupted 3-step sequence.** All syntheses involve preparation of a Grignard reagent in situ from the corresponding aryl halides and subsequent addition to proline derivative, acidic deprotection, and silyl ether formation. Different reagent inputs are provided into otherwise identical digital procedures for execution on an automated synthesis platform, allowing for convenient access to multi-gram quantities of chiral catalysts (S)-Cat-1–3.

In order to further generalize the approach and avoid having to hard-code the list of parameters and the corresponding Blueprint calls, a pattern-matching functionality was implemented for iterations (Fig. 6A). Thus, a Repeat block can iterate over reagents, components, or parameters, and match them by a variety of attributes such as, for example, physical state (for reagents), assigned role (for reagents), type (for components and parameters), among others. Any properties that are defined during initialization can be utilized for pattern matching to yield a generator. In the absence of matched iteration variables, the inputs are resolved normally, i.e. by name. Furthermore, different iteration modalities are accessible through Repeat blocks – standard iteration, equivalent to for each loops in standard programming languages (Fig. 6A, left); partial iteration (Fig. 6A, right top); and flat iteration (Fig. 6A, right bottom), which maintains the state of iterator, rather than reinitializing the iterator. Combination of these iteration modes allows for different behaviors to be achieved. For example, full combinatorial iteration is achieved through combining multiple for each-type Repeat blocks, which could be used in high throughput experimentation (HTE) to screen all possible combinations of variables. Partial combinatorial iteration can be used to enforce a stricter design, where certain parameters are dependent on each other. Flat iteration allows generation of union between combinatorial combinations and flat list of parameters. Therefore, an experiment which uses **Cat-3** to generate multiple batches of Michael addition products can be succinctly encoded within a single Repeat block for multiple repetitions of the same product, or a single nested Repeat block for multiple products.

In order to further optimize and generalize the process, the overall procedure was split into three main sections – reaction set-up and execution, catalyst recycling, and product isolation. Each section was encoded as a standalone Blueprint, which is general and can be applied to different substrates by providing necessary Parameters (Fig. 6B, left). As product isolation via liquid-liquid separation, evaporation, drying, and subsequent transfer to storage is independent of the set-up and execution of the following reaction cycle, certain operations could be parallelized, making the overall process considerably more time-efficient compared to fully linear execution of all unit steps (Fig. 6B, right). The catalyst recovery (Recycling) was therefore scheduled as the main process queue – meaning all other queues wait for operations in the main queue to finish before starting, and additionally have to be finished before the next operation of the main queue is started. This enables a dynamic execution scheduling, where the order of operations is not hard-coded and is not necessarily fixed. Instead, a topological relationship is defined, describing prerequisites for each operation, and specifying points in the synthesis script where the execution of steps can diverge or has to converge. To make use of such functionality, the reaction setup (Reaction) and further workup of the product (Workup) could be scheduled into separate queues and were run in parallel i.e. Workup of Cycle 1 would run in parallel to the Reaction of Cycle 2. Using the previously reported reaction conditions, 4 separate fractions of product **9a** could be obtained with good to excellent yields and stereoselectivities (see Table S2). In our hands, the reaction was observed to be faster in the initial cycles, resulting in poorer diastereoselectivity of the products due to scrambling of the α-chiral center by the catalyst, and slower in later cycles, resulting in decreased conversion, presumably due to cumulative decomposition of the catalyst. It is worth noting that executing otherwise identical procedure in a linear fashion, i.e., without parallelizing work-up and reaction sections, further amplified the catalyst

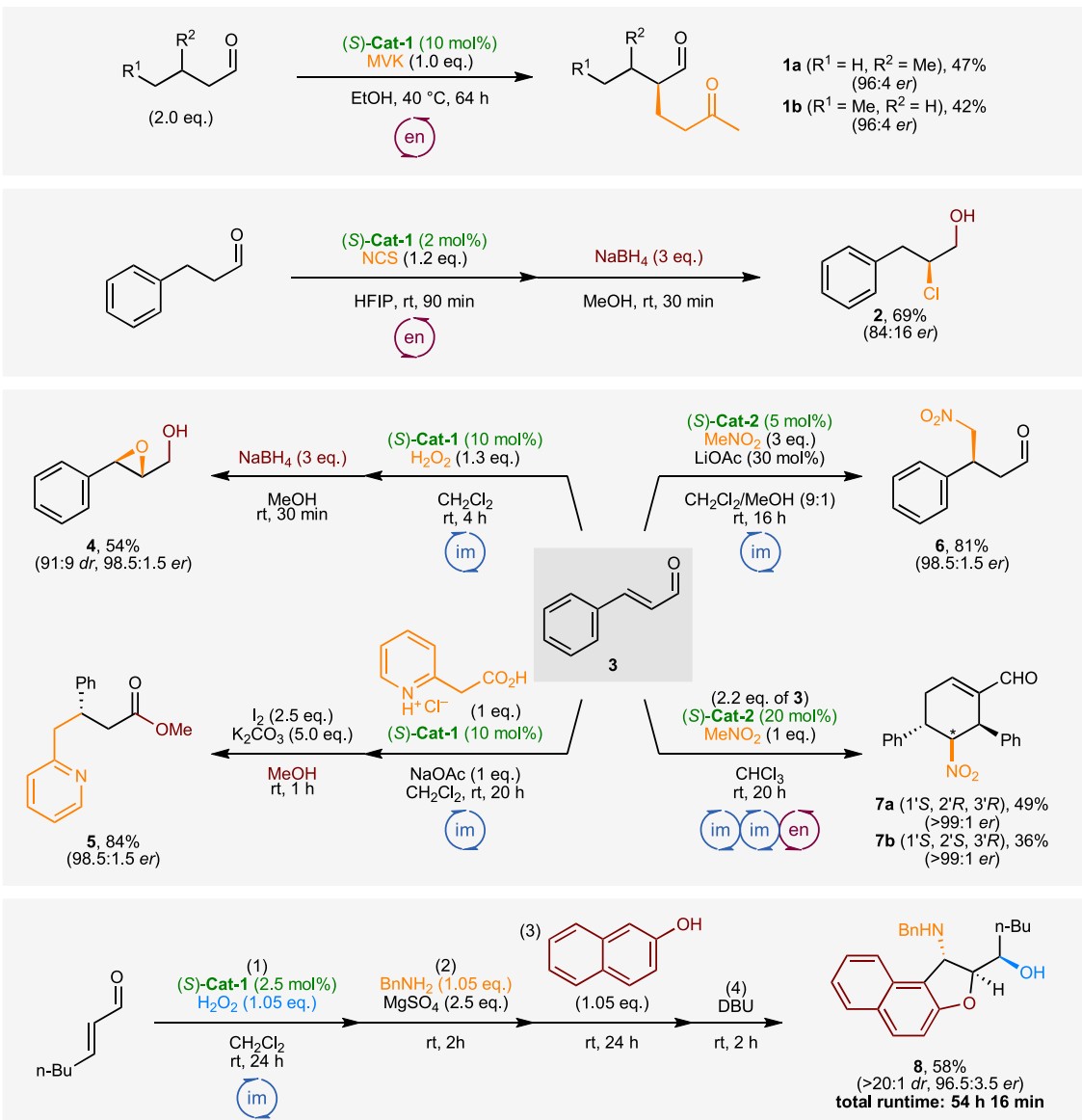

**Fig. 5 | Applications of catalysts (S)-Cat-1 and (S)-Cat-2 in synthesis of enantiomerically enriched molecules.** α-functionalization of aldehydes via enamine intermediates (en) affords Michael addition products **1a,b** and α-chlorinated **2**. Functionalization of unsaturated aldehydes (**3**) via imininium intermediates (im) affords epoxidation product **4**, decarboxalytive Michael addition product **5**, Henry product **6**, and double Henry-aldol domino cascade products **7a,b**. Longer cascade sequences utilizing in situ derivatization of epoxidation product allows rapid construction of complex scaffolds such as **8** with multiple stereogenic centers in a fully automated synthetic procedure. MVK methyl vinyl ketone, NCS N-chlorosuccinimide, DBU 1,8-diazabicyclo[5.4.0]undec-7-ene.

decomposition and the efficacy of the subsequent cycles rapidly dropped.

We, therefore, made use of Parameters to enable easy modification of crucial reaction parameters, such as reaction time, for specific experiments without otherwise changing the general protocol. Each iteration receives a reaction time parameter, and the reaction time was decreased for Cycle 1, and increased for Cycle 4 compared to the baseline literature conditions. The combination of Parameters, Blueprints, and iteration allows for expressing more complex experimental setup in a concise manner, and the same reaction sequence could still be encoded within a single Repeat block by iterating over pre-defined reaction times.

With the improved protocol, 4 fractions of the product **9a** could be obtained with improved yield and stereoselectivity (Fig. 6C). Encouraged by the efficiency of the autonomous catalyst recycling process, we sought to apply it to the synthesis of a small library of compounds, recycling and reusing the catalyst in the process. Without

any modifications to the previously encoded reaction Blueprints, we were able to execute the same process using a combination of 2 aldehydes and 2 nitrostyrenes, to afford products **9a–d**. In this case, a process using the recyclable catalyst in four separate reactions with four different combinations of starting materials and reaction times was encoded in a single nested Repeat block, utilizing previously defined Blueprints. This advanced experimental setup was conveniently expressed using χDL syntax and the interpreter combined the required discrete parameters (reaction times, available hardware) with all possible combinations of available reagent pairs to generate a list of experiments to perform. Execution of such Repeat block afforded the target products **9a–d** in a single automated run.

It is notable that the time required to execute these procedures is highly consistent — both the process affording 4 batches of the product **9a**, and the process afford 4 different products **9a–d** were executed with essentially identical run times. Importantly, both procedures were executed over nearly 28 hours of continuous

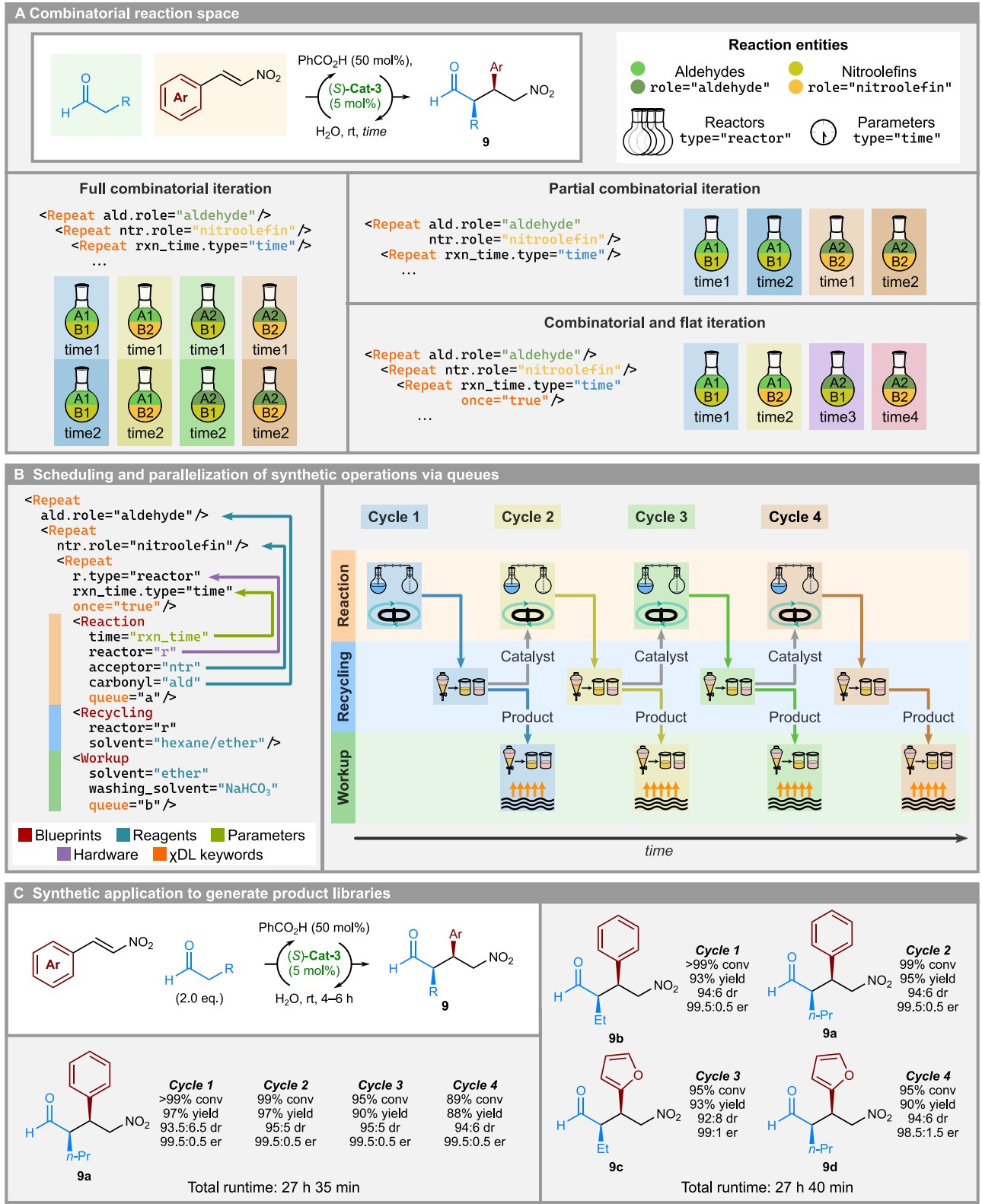

**Fig. 6 | Synthetic applications utilizing χDL iterations, blueprints, and parallelization. A** Iterations over reagents, hardware, and parameters are possible through Repeat blocks. Different iteration modes, including combinatorial and flat iteration, can be combined to generate different experimental designs. All possible arrangements are described succinctly using nested Repeat blocks. **B** Parallelization through queues allows multiple independent operations to be run at the same time, as well as to describe prerequisites for each operation (arrows indicate control flow of the process). Pictured is a workflow for recycling an organocatalyst in an asymmetric Michael addition reaction. **C** Automated synthesis of several batches of Michael addition products with catalyst recycling. Combined use of iterations, blueprints, and parallelization enables multiple batches of the same product (left) or a small library of products (right) to be generated with a single dosage of the catalyst.

operation, with material manipulations, such as reagent additions, reaction mixture transfers, and liquid-liquid separation workups, being performed throughout the whole period. Such synthetic procedures are inherently inconvenient for human chemists and, in fact, completely impossible to execute within standard working hours. Furthermore, among the reactions described above, all of the optimal literature reported procedures utilized reaction times that were either very short (less than 4 hours) or the reactions were generally left overnight (loosely defined, typically 16–20 h), and very few protocols in current literature explore and utilize the intermediary reaction times during optimization. Such disparity is unsurprising as the current literature contains procedures developed by human chemists to be executed by other human chemists, and therefore omits a large portion of reaction conditions from reaction development, as they would not be practical to execute. Further optimization of these and other synthetic protocols towards automation, and the design of digital protocols that benefit from parallelization, and other time saving measures, would undoubtedly unlock still untapped potential of modern synthetic chemistry.

The development of a universal abstraction and a portable, hardware-independent programming language was a significant step in the progress of digital chemistry by associating all common synthesis steps with unambiguous definitions for execution on any robotic platform. We have now taken the next logical step by adapting concepts from high-level programming languages to digital chemistry and demonstrating their ability to open up a wider range of chemistry experiments – not just straightforward linear sequences of operations – to chemical automation. Rather than a direct translation of mainstream programming language constructs, our implementation is tailored to operate on reagents, hardware, reactors, and reaction as the fundamental entities used in digital chemistry.

We were able to use the new language constructs – blueprints, iteration, and queues – to encode a complex synthetic cascade intuitively whilst guarantying correct parallel execution for maximum efficiency. Manual sequencing of this ambitious experimental design would have been fragile and error-prone, leading to inefficient operations that are hard to understand and generalize. We used this added functionality to automate the preparation of chiral organocatalysts and their subsequent application in stereoselective synthesis as a showcase of their expressive power and practical value. Additionally, benefits of automation were exploited for automated recycling and reuse of a water-soluble organocatalyst, and the process was adapted to rapid generation of small libraries of closely related products.

## Methods

Supplementary Information is linked to the online version of the paper and can be found here. Data includes supplementary information, synthesis scripts for and NMR data of all the products.

## Data availability

The authors declare that the data supporting the findings of this study are available within the paper and its Supplementary Information files. Should any raw data files be needed in another format, they are available from the corresponding author upon request. A collection of all χDL (.xdl), graph (.json), and the resulting compiled χDL (.xdlexe) synthesis files for all performed automated syntheses of this study are provided with this paper (Data S1).

## Code availability

The authors declare that the Python package used to execute these automated synthesis files is available from https://gitlab.com/croningroup/chemputer/xdl/-/tree/v2.0.0, further information is available from the corresponding author upon request.

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

## Acknowledgements
We would like to acknowledge financial support from the EPSRC (grant nos. EP/L023652/1, EP/R01308X/1, EP/S019472/1, and EP/P00153X/1), ERC (project 670467 SMART-POM), and FWF (graduate program MolTag W1232, C.K.-F.).

## Author contributions
L.C. conceived the concept and the project idea and supervised the project. M.Š. configured the automated synthesis platform, designed, and performed the experiments. C.K.-F. implemented new software features and helped with performing the experiments. H.M.M. designed the new language features. E.C. and H.M.M. implemented new language features. M.Š. wrote the manuscript with input from all authors.

## Competing interests
The authors declare no competing interests.
