## [Transparent Peer Review file · Nature Communications]

Reaction Blueprints and Logical Control Flow for Parallelized Chiral Synthesis in the Chemputer

Corresponding Author: Professor Leroy Cronin

Version 0:

Reviewer comments:

Reviewer #1

(Remarks to the Author)

Cronin et al. present an improved Chemputer for automated multi-step synthesis, demonstrating the synthesis of diarylprolinol organocatalysts and their use in automated organic synthesis, including recycling and reuse. The major scientific impact of this study is the platform's ability to complete multistep synthesis continuously without the limitation of human work hours. However, the paper shows only a minor improvement from their previous study with a single case study, which may not meet the standards of Nature Communications.

Introduction:

1) The introduction should be rewritten to focus on the motivation and novelty of this study compared to the previous publication. The authors should address:

- a) The differences between this study and the previous one.
- b) The rationale for selecting this specific catalyst and its application.
- c) The challenges of implementing multi-step synthesis in the Chemputer.

2) Paragraph 2, which discusses biosynthesis, should be removed as it is not relevant to the focus of this paper.

3) Paragraph 3 should be replaced with the first paragraph of the Results and Discussion section, which provides a good introduction to the catalysts synthesized in this study.

Results & Discussion:

1) Provide a more detailed explanation of the new functionalities of χ DL, including blueprints, iteration, and parallel execution. A comparison with the previous version of the Chemputer would be beneficial.

2) Clarify the optimization process mentioned on page 9. Was it expert-based or automated? Discuss the potential for incorporating advanced optimization techniques such as Bayesian optimization or reinforcement learning.

3) Discuss any improvements in the quality of the crude and purified products compared to human-operated experiments.

4) Determine the focus of the paper. If it is about the specific reaction, it may be more suitable for a chemistry journal like JACS or Angewandte. If it is about the update of the Chemputer, expand the reaction scope to demonstrate its versatility.

Minor issue:

Clarify the term "ChemPU" in Figure 4 and ensure consistency in terminology.

To summarize, the manuscript presents significant advancements in automated chemical synthesis. However, to meet the standards of Nature Communications, the authors should expand the scope of reactions, provide more detailed explanations of new functionalities, and clarify the optimization process.

Reviewer #2

(Remarks to the Author)

In harmony with the musings that authors shared the area of organocatalysis is well over couple of decades old, I wish to say that a lot of domain knowledge is already available. For textbook examples of the sort described in the manuscript relating to diarylprolinols (Fig. 3) come with prior awareness of various controllable (reagents, their proportion, time, temperature) and uncontrollable factors. In other words, one deals with reasonably known set of (challenging?) parameters.

Whatever this old wine in new bottle workflow could offer would unfortunately remain 'obsolete' as I don't see much recent activities (a simple searching around revealed articles like 'after the golden age of organocatalysis' and a recent Noble prize in this area). Of course, one could argue that choosing an older reaction class would be a safe bet at proof of concept stage. I wonder with all the sophistication (or lack of it!) what do one get, over and above the two-decade-old accrued knowledge. I gather that some of these catalysts are commercially available. The results are certainly not of tour de force kind, but a good culmination of authors experience and an extension their pioneering work on automation.

I don't see much innovation in demonstrating whether the technique can offer valuable improvements for reactions of current interest, or even in the case of a new and emerging class of reaction or new directions to existing (experimental procedure) practices.

In summary, I don't see this manuscript living up to the desirable levels of a Nat.Comm. article.

Minor comments: blueprint details (Figs. 3 and 6) are at best a distraction in its present format. There is no compelling reason to give it a 'programming' flavor rather than just stating the facts using a simple flowchart representation.

Citations are handpicked to reflect the so-called 'top-tier' journals, thus nearly failing to list out important works in the said domain.

Reviewer #3

(Remarks to the Author)

The authors introduce a pioneering method in universal robotic chemistry. They demonstrate its efficacy through the generation of a chiral catalyst and its subsequent utilization across diverse organic transformations. I think this work is quite interesting and can be accepted after addressing some questions below.

1. The work carried out in this manuscript can be considered as the extension of their previous work, using the same platform, i.e. Chemputer. Given that the synthesis platform has been detailed in several papers, what are the primary innovations and advancements in both its code and hardware compared to previously reported systems? Additionally, how efficient is this new system, in terms of time or human intervention required?
2. The authors mentioned 'the final catalysts (S)-Cat-1-3 in yields comparable to those achieved by an expert chemist following the same procedure manually', I would like to see the actual yield values for both methods and understand the source of any differences.
3. As a human chemist, who can easily recognize any unusual phenomena during the execution, while how can one track where the issues come from if using this continuous and fully autonomous system, especially for a multi-step reaction?
4. I wonder if the system would also benefit process development. For example, it mentioned, 'Replacing the acid used in the deprotection to hydrogen chloride resulted in clean deprotection.' Does this change require any manual experiments, or can it be optimized using a Chemputer?

Version 1:

Reviewer comments:

Reviewer #1

(Remarks to the Author)

The authors have solved this issues I raised in the original manuscript. The paper can be accepted for publication now.

Reviewer #3

(Remarks to the Author)

I think that the authors have addressed my concerns in the revised version of the manuscript. Therefore, I have no further comments.

We thank the reviewers for taking the time to provide thoughtful feedback and their recognition of the importance of this work. Below we detail our response to their feedback and queries.

Reviewer #1 (Remarks to the Author):

Cronin et al. present an improved Chemputer for automated multi-step synthesis, demonstrating the synthesis of diarylprolinol organocatalysts and their use in automated organic synthesis, including recycling and reuse. The major scientific impact of this study is the platform's ability to complete multistep synthesis continuously without the limitation of human work hours.

We are glad the reviewer finds the study impactful. We would like to clarify that although achieving multistep synthesis without limitation of human work hours is one of the most obvious and tangible benefits of this work for an organic chemist, there are in fact a number of other areas where the present work provides substantial improvements. The combination of iterations, blueprints, and parameters enables encoding of more complex experiments in a general form. Such universal encoding is a crucial prerequisite for developing more complex automated procedures, and standardizing the field of digital chemistry.

However, the paper shows only a minor improvement from their previous study with a single case study, which may not meet the standards of Nature Communications.

Although we have chosen to target a specific area of chemistry to demonstrate the concepts, the principles presented in the manuscript are general to a wide range of chemistries and it should not be viewed as a one-off case study. Our main goal was to develop new functionalities for a high-level chemical programming language, and also apply them to chemistries that pose certain challenges which can be conveniently addressed by expanding the capabilities of chemical programming language. It is worth noting that solving any individual challenge as a one-off problem might be relatively trivial, however, to do so in a generally applicable manner, such as by using a high-level chemical programming language is significantly more complex. The principles presented in this work are not limited to the chemistry used to illustrate said concepts.

Introduction:

1) The introduction should be rewritten to focus on the motivation and novelty of this study compared to the previous publication. The authors should address:
a) The differences between this study and the previous one.

We thank the reviewer for this comment. We agree that the main novelty of this study may not have been communicated clearly.

We have rewritten the abstract and the introduction to better highlight the main motivation behind the study – namely, the fact that digital chemistry approaches, including our previous studies, up to now have relied on linear, mostly verbatim conversion of literature procedures. This does not harness the true potential of digitisation, and concepts that are well-established in conventional programming languages can be applied to the way digital synthesis is captured. Although such features are accessible if synthesis script is written in a standard programming language, to do so in an abstract, generalisable, human-readable format, such as XDL, is a major deviation from *status quo*.

We have reworked Figure 2 to better illustrate the main difference between this study and previous ones. Current work presents a structured chemical programming approach, and implements features found in conventional programming languages, such as variables, functions, iteration, and execution scheduling. It does so while maintaining the high-level chemical programming framework, and remains compatible with previous work.

b) The rationale for selecting this specific catalyst and its application.

We have modified the introduction to highlight that synthesis of chiral building blocks is often a major bottleneck, even for other automation approaches. Such work is tedious and time-consuming, often includes synthesis of chiral catalysts before their use in synthesis of the required building blocks. We have chosen diarylprolinol derivatives as a class of catalysts that illustrates this concept, which are often made in-house on multi-gram scale. Such repetitive work would ideally be relegated to automated synthesis platforms.

We agree that the rationale for selecting this specific catalyst has not been explained in depth, however, from reading all reviewers comments we concluded that the main focus of this study was not communicated clearly – namely, both Reviewer 1 & Reviewer 2 questioned the novelty and significance of the chemistry shown. We would like to reiterate that the chemistry presented is not meant to be considered somehow novel or unique. We did see it as a useful tool in the vast toolbox of synthetic chemists, yet one that poses certain challenges in its application.

We sought to develop previously reported chemical description language (XDL) into a more robust structured chemical programming language. The particular class of catalysts and reactions for their use was chosen to illustrate how structured programming can enable convenient automation

The choice of reactions can be considered somewhat arbitrary, however, we believe that synthesis of this particular class of catalysts and their use in single-, and multi-step reactions, as well as complex time-sensitive recycling experiments illustrate these concepts well.

c) The challenges of implementing multi-step synthesis in the Chemputer.

It is worth noting that we mainly targeted generalized multi-step synthesis, that is, when small libraries of related compounds can be prepared with well defined changes in input reagents or parameters but otherwise follow a generic sequence of operations. This was one of the main inspirations for the development of Blueprints – standard synthetic protocols with well defined parametrization.

Additionally, for multi-step synthesis or complex multi-reaction experiment, there is a linear chain of dependencies. Now, with these chains broken into Blueprints, there is scope of using an optimised structure flow (see Fig 6 and related discussion), fault recovery, isolated execution of specific sections. We have implemented execution scheduling *via* process queues, which provides a convenient way of describing non-linear chain of dependencies.

2) Paragraph 2, which discusses biosynthesis, should be removed as it is not relevant to the focus of this paper.

We have removed the discussion about biosynthesis and condensed the discussion into the key point we were trying to convey – iteration is important in biosynthesis, solid phase peptide synthesis, and increasingly in synthetic chemistry.

The main text now reads:

Complimentary to the advances in automation, a growing number of synthetic methodologies targeted directly towards automated synthesis are being developed. Particularly, several novel iterative approaches have been recently reported, which enable a small set of operations to be developed and optimised for automated execution

3) Paragraph 3 should be replaced with the first paragraph of the Results and Discussion section, which provides a good introduction to the catalysts synthesized in this study.

Paragraph 3 has been condensed and combined with the main point of the following paragraphs – even for iterative approaches, preparation of building blocks and derivatisation at the end of iterative sequence is needed and relies on conventional chemical transformations.

We have significantly condensed the discussion pertaining to specific chemistries in the introduction and focused more on the current challenges in the state-of-the-art automation and the developments to XDL we have made to address these challenges.

Results & Discussion:

1) Provide a more detailed explanation of the new functionalities of χ DL, including blueprints, iteration, and parallel execution. A comparison with the previous version of the Chemputer would be beneficial.

We have rewritten the introduction to detail the new functionalities and the motivation for their inclusion. Figure 2 has been reworked to show the comparison with the previous approaches to the chemical programming. We would like to stress a key point – *Chemputer* as a synthesis platform has not changed, and all improvements come from new developments to the XDL. The key feature and goal of XDL is to be a universal chemical description language, that can be executed on any suitable automation platform.

A detailed explanation for the motivation and working principles of Blueprints can be found in Figure 3 and accompanying discussion.

Figure 6 has been reworked to better illustrate the parameter, blueprint, iteration, pattern matching and parallel execution concepts. An expanded explanation added to the discussion of Figure 6.

We hope these changes clarify our motivation for implementing new features, and highlights their importance not just for the chemistry shown, but in broader context as well.

2) Clarify the optimization process mentioned on page 9. Was it expert-based or automated? Discuss the potential for incorporating advanced optimization techniques such as Bayesian optimization or reinforcement learning.

SM section for synthesis of **(S)-Cat-2** contains info on the optimisation process. To clarify, the optimisation only changed the acid used and the concentration in the case of HCl. It was a quick screen after a specific problem was identified with previous procedure, rather than an in-depth campaign looking at a variety of parameters. In this case, automated ML or related approaches are not required, and the decision was expert-based.

We agree with the reviewer that combination of the techniques presented in this work with advanced optimization techniques such as Bayesian optimization or reinforcement learning would be incredibly powerful but it is outside the scope and focus of this work.

3) Discuss any improvements in the quality of the crude and purified products compared to human-operated experiments.

There was no expectation to produce products of higher quality. The biggest benefits of the automated workflow are convenience, and option for uninterrupted operation. This can lead to lower time-to-completion, e.g. catalyst synthesis requires at least 2-3 human working days cf. continuous 36 hours of operation with no human intervention for the automated platform. Catalyst recycling experiments requires continuous operation for 24+ hours in 5 hour intervals – this is simply impossible within normal human working hours. This, of course, can lead to improved quality of products obtained in automated synthesis compared to human-operated experiments if human chemist is unable to perform operations at optimal times.

4) Determine the focus of the paper. If it is about the specific reaction, it may be more suitable for

a chemistry journal like JACS or Angewandte. If it is about the update of the Chemputer, expand the reaction scope to demonstrate its versatility.

We are not presenting any new chemistry or focusing on a specific reaction. The main goal was to present a new way of digitally representing synthetic procedures by utilizing the principles of computer science/programming, and chemistry was chosen to showcase that such features significantly improve the convenience and/or enable workflows that would otherwise have been impossible.

We would like to stress again that advancements in the chemical description language (XDL) presented within this work are of high importance for modernization of chemistry. The new features enable convenient and standardized description of complex experiments, such as those requiring iterations, combinatorial combinations, scheduling/priority queues of operations, etc.

We have shown relatively broad applicability of the approaches presented by synthesizing multiple catalysts using the blueprints, using them in a range of organocatalyzed reactions to general chiral building blocks, and exemplified the key principle of iterations by recycling a catalyst with a) the same substrate; b) with varied substrates to generate a library of products. We believe the general strategy and the underlying principles have been sufficiently demonstrated and expanding the scope would not add any further proof to the concept other than producing product X or Y.

Overall, the abstract, introduction, and the results sections have been rewritten to highlight the focus of the paper better, in particular, the motivation and importance of structured chemical programming. We hope that the reviewer finds the message a lot clearer in the revised manuscript.

Minor issue:

Clarify the term "ChemPU" in Figure 4 and ensure consistency in terminology.

The usage of terms has been made consistent – we refer to the automation platform used in the study as *Chemputer*.

To summarize, the manuscript presents significant advancements in automated chemical synthesis. However, to meet the standards of Nature Communications, the authors should expand the scope of reactions, provide more detailed explanations of new functionalities, and clarify the optimization process.

We are glad the reviewer finds the work presented a significant advancement in automated chemical synthesis. We thank the reviewer for their constructive feedback and insightful comments which have helped us improve the manuscript.

Reviewer #2 (Remarks to the Author):

In harmony with the musings that authors shared the area of organocatalysis is well over couple of decades old, I wish to say that a lot of domain knowledge is already available. For textbook examples of the sort described in the manuscript relating to diarylprolinols (Fig. 3) come with prior awareness of various controllable (reagents, their proportion, time, temperature) and uncontrollable factors. In other words, one deals with reasonably known set of (challenging?) parameters.

We agree that there are a number of known and unknown challenging parameters, which present an obstacle for generalized automated synthesis. We present a novel way of representing such parameters in order to digitally capture the synthetic procedures in a universal format. We would like to emphasize that we are not presenting the chemistry itself as new – we are using it to

showcase convenience of digitally-oriented approaches and their importance for automated synthesis.

Whatever this old wine in new bottle workflow could offer would unfortunately remain 'obsolete' as I don't see much recent activities (a simple searching around revealed articles like 'after the golden age of organocatalysis' and a recent Noble prize in this area).

We disagree that organocatalysis is obsolete – there are a plethora of research works being published in the area of organocatalysis, and its combination with newer research areas e.g. photoredox chemistry (*Nat. Catal.* **2023**, 6, 332–338, 10.1038/s41929-023-00939-y), electrochemistry (10.1002/anie.202401361), and is being further reinvigorating by using modern digital chemistry techniques, such as ML (10.26434/chemrxiv-2024-xfdn8).

In either case, the focus of this work is not any specific chemistry but the underlying principles of digital chemistry and addressing the current limitations for automated synthesis. The organocatalysis was chosen as a case study to showcase the workflow advances. The exact same principles apply to other areas of synthetic chemistry and we have previously shown the breadth of chemistries that are compatible with our platform and how the syntheses can be encoded in chemical description language (*Science*, **2022**, 377, 172-180, 10.1126/science.abo0058).

We have rewritten the introduction to explain our motivation for developments in chemical programming better, and condensed the chemistry discussion to help convey the message better,

Of course, one could argue that choosing an older reaction class would be a safe bet at proof of concept stage. I wonder with all the sophistication (or lack of it!) what do one get, over and above the two-decade-old accrued knowledge. I gather that some of these catalysts are commercially available. The results are certainly not of tour de force kind, but a good culmination of authors experience and an extension their pioneering work on automation.

We agree with the reviewer that the chemistry presented within is relatively established but presents multiple challenges, such as time-sensitive operations (e.g. *er* for chlorination, *dr* for Michael in fig 6.) For recycling experiments, the catalyst loses efficacy over time and requires operations at specific time intervals for success. In fact, the original procedure for reaction shown in Figure 6 is impossible to execute within standard human working hours.

The reviewer is correct in stating that *some of the catalysts* are commercially available, however, many research labs choose to make them, for multiple reasons, such as, exploring reactivity of non-commercially available catalysts, and higher quality of freshly made and/or well-stored catalysts. Certainly, very few research chemists would be excited by the prospect of being tasked to make some catalyst for the lab, which is precisely where we believe automated synthesis generates value for research chemists – replacing tedious and repetitive synthetic work.

We believe the present work significantly expands the scope of capabilities of the automated synthesis and allows for convenient description of complex synthetic procedures, which can be easily reused.

I don't see much innovation in demonstrating whether the technique can offer valuable improvements for reactions of current interest, or even in the case of a new and emerging class of reaction or new directions to existing (experimental procedure) practices.

Although we have chosen to target a specific area of chemistry to demonstrate the concepts, the principles presented in the manuscript are general to a wide range of chemistries and it should not be viewed as a one-off case study. Our main goal was to develop new functionalities for a high-level chemical programming language, and also apply them to chemistries that pose certain

challenges which can be conveniently addressed by expanding the capabilities of chemical programming language.

In summary, I don't see this manuscript living up to the desirable levels of a Nat.Comm. article. Minor comments: blueprint details (Figs. 3 and 6) are at best a distraction in its present format. There is no compelling reason to give it a 'programming' flavor rather than just stating the facts using a simple flowchart representation.

We disagree, in fact, blueprints, iterations, and parallel execution are the key underlying principles of the present work, and have been underutilized in the state-of-the-art up to now. Although a growing number of laboratories are starting to utilize automated synthesis in some form (HTE, custom synthesizers, etc.), a general, digital description of synthetic protocols has not been in the focus of research. We believe that standardisation of digital synthetic protocols is crucial for modern chemistry, and to truly modernize synthetic chemistry, principles of programming need to be incorporated into the descriptions of synthetic procedures, particularly so for automated platforms. This not only makes the encoding of procedures significantly more convenient, but also unlocks functionality that was not available previously.

We appreciate that perhaps this message has not been communicated clearly enough, and have highlighted this better by rewriting abstract, introduction, reworking Figures 2 and 6. We hope with these changes the reviewer is able to appreciate that new features described in this work are not just 'programming flavor' and rather cornerstone aspects of modern digital chemistry.

Citations are handpicked to reflect the so-called 'top-tier' journals, thus nearly failing to list out important works in the said domain.

We believe the citations cover a range of state-of-the-art in automation, digital chemistry and relevant advances in chiral organocatalysis. The synthesis protocols used in the applications of catalysts in synthesis were cited as their original publications.

We would be happy to incorporate any missing citations of the important works from said domains.

Reviewer #3 (Remarks to the Author):

The authors introduce a pioneering method in universal robotic chemistry. They demonstrate its efficacy through the generation of a chiral catalyst and its subsequent utilization across diverse organic transformations. I think this work is quite interesting and can be accepted after addressing some questions below.

We are glad the reviewer has found our work interesting.

1. The work carried out in this manuscript can be considered as the extension of their previous work, using the same platform, i.e. Chemputer. Given that the synthesis platform has been detailed in several papers, what are the primary innovations and advancements in both its code and hardware compared to previously reported systems?

The hardware of the synthesis platform has remained unchanged and the primary improvements come from new software features, which enable convenient digital capture and execution of more complex synthetic experiments.

We have rewritten the introduction and reworked Figure 2 to better explain the motivation and to illustrate the main differences between this study and previous ones. Current work presents a structured chemical programming approach, and implements features found in conventional programming languages, such as variables, functions, iteration, and execution scheduling. It does so while maintaining the high-level chemical programming framework, and remains

compatible with previous work. More in-depth discussion of these features can be found in Figure 3 and its accompanying text, and reworked Figure 6 and its expanded accompanying text.

Additionally, how efficient is this new system, in terms of time or human intervention required?

Up to now we have not shown any multi-step procedures executed without human intervention. The new features presented in this study enabled us to conveniently capture the required parameters, and execute several related multi-step procedures. No intervention was required during the multi-step reaction sequence, and only minimal intervention (e.g. loading a different substrate) was needed to repeat the multi-step sequence to produce a different product of the same class. Likewise, no intervention is needed in the recycling study during the run (4 cycles), and the multi-reaction process could be repeated with different substrates by simply swapping the starting materials connected to the platform.

Parallel execution not only provides time saving benefits (e.g. reaction shown in Figure 6 to produce **9a** performed under the exact same conditions in “linear” fashion took ~32 hours cf. ~27.5 hours with parallel execution), it can also be crucial to the success of the reaction. For example, in the catalyst recycling study, running the reaction without any parallelisation, leads to significant catalyst decomposition and affords the products in severely diminished yields. (the results are detailed in SM section for synthesis of **9a**). Parallel execution was critical to perform such procedure successfully in automation.

It is worth noting that exact time/efficiency bonus depends on actual reaction/experiment in question. The features presented in this study merely enable such benefits by providing a convenient framework for encoding complex experiments in a high-level structured chemical programming language.

We have expanded explanation for Figure 6 to include more detail on execution scheduling and why it was crucial in this particular case study – linear execution both took longer, and lead to significant erosion of stereocontrol and conversion .

2. The authors mentioned ‘the final catalysts (S)-Cat-1-3 in yields comparable to those achieved by an expert chemist following the same procedure manually’, I would like to see the actual yield values for both methods and understand the source of any differences.

The sentence in the main text has been reworded to read:

Excitingly, the uninterrupted three-step sequence executed in automation afforded the final catalysts in yields comparable to those achieved by an expert chemist following the same procedure manually, for example, manual synthesis of rac-Cat-2 afforded the catalyst in 83% yield (cf. 77% for automated synthesis of (S)-Cat-2).

We consider the yield to be within experimental variation and/or variation in starting material quality - the manual synthesis was performed on a 1:1 ratio of enantiomers of N-Boc-Pro-OMe.

3. As a human chemist, who can easily recognize any unusual phenomena during the execution, while how can one track where the issues come from if using this continuous and fully autonomous system, especially for a multi-step reaction?

The reviewer is correct in that this is a major challenge in the current state-of-the-art. Our current system utilizes video feedback, which a human operator can access to inspect the platform remotely at any point during execution or review afterwards.

We have not used any real-time monitoring within this work but have reported monitoring/dynamic approaches in previous communications, e.g. *Nat. Commun.*, **2024**, *15*,

(doi: 10.1038/s41467-024-45444-3), *Nat. Chem.*, **2022**, 14, 1311–1318
<https://doi.org/10.1038/s41557-022-01016-w>.

We are actively working on expanding the scope and applications of such features but full integration is a monumental challenge and is outside the scope of this work.

It is worth noting that the main idea of blueprints is to capture an optimized, reliable procedure, that can be easily repeated as needed. This requires some initial process development work but produces significantly more reliable automated workflows.

4. I wonder if the system would also benefit process development. For example, it mentioned, 'Replacing the acid used in the deprotection to hydrogen chloride resulted in clean deprotection. ' Does this change require any manual experiments, or can it be optimized using a Chemputer?

SM section for synthesis of **(S)-Cat-2** contains info on the optimisation process used for the deprotection step. In this case, the experiments were performed manually but it could have been performed using the Chemputer, or any other automated platform.

We agree with the reviewer that this approach could be very beneficial to process development, where minor changes to the synthesis protocol are evaluated. We believe that capturing the workflows as blueprints facilitates such process developments, and was one of the inspirations for introducing blueprint functionality. Combined with the iterations (Figure 6), complex condition screening experiments can be succinctly described. In this work we focused on preparative chemistry to access chiral catalysts and chiral building blocks, but the reviewer is correct in identifying other potential applications for the principles presented.